# Transcriptome Analysis of Intracellular Amastigotes of Clinical *Leishmania infantum* Lines from Therapeutic Failure Patients after Infection of Human Macrophages

**DOI:** 10.3390/microorganisms10071304

**Published:** 2022-06-27

**Authors:** Raquel García-Hernández, Ana Perea-Martínez, José Ignacio Manzano, Laura C. Terrón-Camero, Eduardo Andrés-León, Francisco Gamarro

**Affiliations:** Instituto de Parasitología y Biomedicina “López-Neyra”, IPBLN-CSIC, Parque Tecnológico de Ciencias de la Salud, Avda. del Conocimiento 17, Armilla, 18016 Granada, Spain; raquelg@ipb.csic.es (R.G.-H.); anapereama@ipb.csic.es (A.P.-M.); nacho.manzano@ipb.csic.es (J.I.M.); lauteca@ipb.csic.es (L.C.T.-C.); eduardo.andres@csic.es (E.A.-L.)

**Keywords:** *Leishmania infantum*, intracellular amastigotes, drug-resistant parasites, therapeutic failure clinical parasites, host infection, transcriptomic analysis, RNA-seq, differential gene expression

## Abstract

Leishmaniasis is considered to be one of the most neglected tropical diseases affecting humans and animals around the world. Due to the absence of an effective vaccine, current treatment is based on chemotherapy. However, the continuous appearance of drug resistance and therapeutic failure (TF) lead to an early obsolescence of treatments. Identification of the factors that contribute to TF and drug resistance in leishmaniasis will constitute a useful tool for establishing future strategies to control this disease. In this manuscript, we evaluated the transcriptomic changes in the intracellular amastigotes of the *Leishmania infantum* parasites isolated from patients with leishmaniasis and TF at 96 h post-infection of THP-1 cells. The adaptation of the parasites to their new environment leads to expression alterations in the genes involved mainly in the transport through cell membranes, energy and redox metabolism, and detoxification. Specifically, the gene that codes for the prostaglandin f2α synthase seems to be relevant in the pathogenicity and TF since it appears substantially upregulated in all the *L. infantum* lines. Overall, our results show that at the late infection timepoint, the transcriptome of the parasites undergoes significant changes that probably improve the survival of the *Leishmania* lines in the host cells, contributing to the TF phenotype as well as drug therapy evasion.

## 1. Introduction

Leishmaniasis is a disease caused by the protozoan parasite *Leishmania* that involves a wide variety of clinical manifestations, ranging from self-healing cutaneous lesions to visceral disease, the latter of which can be fatal if left untreated. Treatment options vary between regions but are generally limited to only a few first-line drugs, including miltefosine (Mil), paromomycin (PMM), amphotericin B (AmB), and antimonials (Sb). All have limitations in terms of toxicity, efficacy, price, and a significant increase in resistance and therapeutic failure (TF). TF has a multifactorial origin, involving a considerable number of factors concerning (i) the host (immunity or nutritional status), (ii) the parasite (drug resistance, infectivity, parasite localization, accessibility to drugs and coinfection with other pathogens), (iii) the drug (quality, pharmacokinetics), and (iv) the environment (global warming and the expansion of the disease to new geographical areas). Thus, strategies for confronting these drawbacks must include new efficient treatment options [1,2].

Changes at the molecular level in *Leishmania* parasites associated with drug resistance have been described [1], including, among others, decreased drug entry in host cells infected by *Leishmania* lines [3,4,5,6,7,8,9]. Additionally, it has been shown that there is an amplification of the ABC transporter PGPA gene in *Leishmania tarentolae* that leads to a sequestration of the metal–thiol conjugates into vesicular membranes, conferring resistance to arsenites and antimonies [10,11]. Furthermore, an increased drug efflux due to the overexpression of ABC transporters has been described [8,9,12,13,14,15]. However, reports of molecular changes in *Leishmania* lines associated with TF are significantly scarce, and more studies exploring this aspect are necessary [1]. A *Leishmania infantum* genetic marker associated with Mil TF has been associated with the absence of a miltefosine sensitivity locus (MSL); this MSL has been found in the genomes of *L. infantum* and *Leishmania donovani* lines that have a cure rate above 93% [16]. The MSL has been considered to be a potential molecular marker for predicting the efficacy of Mil treatment”. An additional factor that may also affect clinical efficacy and lead to treatment failure is the presence of *Leishmania* RNA virus 1 (LRV-1) in parasites of the *Viannia* subgenus. Recent data have indicated that the virus may subvert the host immune response and affect the clinical response to drugs [17,18].

Knowledge of molecular changes associated with drug resistance and TF could help to increase the efficiency of new therapeutic strategies to be used in patients with leishmaniasis and TF. However, these studies have been undertaken principally in promastigote forms of *Leishmania* lines that are experimentally resistant to the common drugs, with very few studies including the use of clinical drug-resistant isolates and/or clinical isolates from patients with leishmaniasis and TF [19,20,21]. Molecular approaches to characterize the mechanisms of drug resistance or TF in these *Leishmania* lines include microarray studies [22], proteomic analysis [23], and, more recently, RNA-seq [24,25,26,27].

*Leishmania* are parasites whose life cycle requires them to infect and replicate within host cells as intracellular amastigote forms, the most important clinical forms of this organism. Consequently, the use of intracellular amastigotes may be useful for studies directed to know the transcriptomic changes associated with the parasite’s ability to survive, evade the host defenses, and elude the toxic effects of anti-leishmanial drugs. During a host–pathogen interaction, global changes in the gene expression pattern occur both in the host (macrophages) [28,29,30,31,32] and in the infecting pathogens [33,34,35,36,37,38,39,40,41,42].

Recently, based on the dual RNA-seq method, we identified host-specific genes that were modulated after the host macrophages interacted with clinical isolates of *L. infantum* with different levels of drug susceptibility to the anti-leishmanial drugs associated, or not, with TF [28,29]. In this article, we analyzed the transcriptomic changes in these intracellular amastigotes of *L. infantum* parasites at 96 h post-infection of THP-1 cells.

Our results contribute to the identification of genes with an altered expression pattern in several *L. infantum* clinical isolates in the amastigote form after a late infection. The association between these genes and TF is described in this work and expands the knowledge of TF and drug resistance in leishmaniasis.

## 2. Materials and Methods

### 2.1. Chemical Compounds

Triton X-100, 4′,6-diamidino-2-phenylindole dilactate (DAPI), phorbol 12-myristate 13-acetate (PMA), and sodium dodecyl sulfate (SDS) were purchased from Sigma-Aldrich (St. Louis, MO, USA). L-glutamine and penicillin/streptomycin were obtained from Gibco. All chemicals were of the highest quality available.

### 2.2. Culture of Leishmania infantum Lines and THP-1 Cells

We used promastigotes of *L. infantum* lines: (i) JPC-M5 (MCAN/ES/98/LLM-877) (LJPC) is a reference line for genomics sequence and for transcriptomic studies that have been employed in several publications [27,43] (in this way, this line is sensitive to all the drugs tested) [30]; (ii) LEM3323 and LEM5159, an Sb-resistant line and a Mil-resistant line, respectively, isolated from human immunodeficiency virus (HIV) patients with visceral leishmaniasis (VL) [44] from Dr. L. Lachaud, France; (iii) LEM2126, a PMM-resistant line isolated from an HIV patient with nephrotic syndrome [45]; (iv) LLM2070; (v) LLM2165; (vi) LLM2255; and (vii) LLM2221. The last four *L. infantum* lines were isolated from TF HIV patients with VL unsuccessfully treated with liposomal AmB (from the WHO Collaborating Center for Leishmaniasis, Instituto de Salud Carlos III; Dr. F. Javier Moreno). The sensitivity of all the above *Leishmania* lines to the different anti-leishmanial drugs has been recently published [28,29].

All these *L. infantum* lines were grown at 28 °C in an RPMI 1640-modified medium (Invitrogen) supplemented with 10% heat-inactivated fetal bovine serum (hiFBS) (Invitrogen) as described [30]. Human myelomonocytic cells THP-1 were grown at 37 °C and 5% CO_2_ in an RPMI-1640 medium supplemented with 10% hiFBS, 2 mM glutamate, 100 U/mL penicillin, and 100 mg/mL streptomycin as described [46]. The THP-1 cell line was isolated from a patient with acute monocytic leukemia [47], and these macrophage-differentiated-THP-1 cells have been widely used in the scientific community as a suitable host model for infection of *Leishmania* lines [48].

### 2.3. In Vitro Macrophage Infection

THP-1 cells (3 × 10^6^ cells in 25 cm^2^ flasks or 5 × 10^5^ cells/well in 24-well plates) were differentiated to macrophages and infected with different *L. infantum* stationary-phase promastigotes incubated for 72 h in an acid medium plus 10% hiFBS at a macrophage/parasite ratio of 1:10 [28,29]. Infected macrophages were incubated in an RPMI-1640 medium plus 10% hiFBS at 37 °C and 5% CO_2_ for 96 h. Finally, the samples were collected in Qiazol (Qiagen, Hilden, Germany), and total RNA was isolated at the Genomic Unit of GENyO facilities (Granada, Spain), as described later. To determine the percentage of infection and the average number of amastigotes per cell, the macrophages were infected with the same *L. infantum* lines and treated in parallel with the same conditions as those employed for RNA extraction. For microscopy visualization, the cells were fixed for 30 min at 4 °C with 2.5% paraformaldehyde in PBS and permeabilized with 0.1% Triton X-100 in PBS for 30 min. Intracellular parasites were detected by nuclear staining with DAPI (Invitrogen, Carlsbad, CA, USA). We observed a percentage of infection of up to 76% in the *L. infantum* lines, with a number of amastigotes per cell of up to 11.

### 2.4. RNA Isolation and cDNA Library Preparation

The protocols used for the RNA isolation of the samples and the generation of cDNA libraries were described previously [28,29]. Briefly, eight different group of samples, defined as LEM2126, LEM3323, LEM5159, LLM2221, LLM2255, LLM2165, LLM2070, and LJPC, with three independent biological replicate experiments each, were mRNA-extracted using a QIAamp RNA miRNeasy Micro Kit (Qiagen/Qiacube), followed by treatment with a DNase kit. RNA integrity was evaluated using an Agilent 2100 Bioanalyzer system with an RNA 6000 Nano Lab Chip kit (Agilent Technologies). Poly(A)-enriched cDNA libraries were generated using a TruSeq Stranded mRNA kit (Illumina). All these analyses and sequencing were performed at GENyO facilities (Granada, Spain).

### 2.5. RNA-seq Data Generation, Pre-Processing

The RNA-seq protocol carried out in this manuscript was previously described [28,29]. In summary, a next-generation sequencing run for whole transcriptome sequencing was performed using a paired-end 2 × 75 nt library on a NextSeq 500/550 Illumina sequencer and using a high-output Kit v2.5 (150 cycles), which generated an average of 18.9 million reads per sample. Raw data were generated for each of the libraries by the genomic core service at GENyO (Granada, Spain). The RNA-seq data are available at NCBI Short Read Archive (SRA) under accession number PRJNA836366.

### 2.6. Data Analysis

The transcriptomic samples were analyzed using the miARma-Seq pipeline [49,50]. This software performs automatically all the required steps to obtain differentially expressed genes (DEGs) from raw data files. Firstly, fastq files were evaluated using the FastQC software to analyze the quality of the reads [51]. On average, 91% of the reads have a mean quality over Q30, and no adapter sequence was found. Subsequently, after filtering sequences by quality (lower than Q30) and homogenizing the number of reads per sample using the Seqtk software [52], miARma-Seq aligns all sequences using HISAT2 [53]. With this aim, *Leishmania_infantum*: LinfantumJPCM5 was used as the reference genome from the TriTrypDB version 38 database. The average percentage of aligned reads against this reference genome was 20.36% (the remaining 60.30% aligned against the human genome). After that, the featureCounts software [52] was used to assign sequence reads with a minimum mapping quality score over 10 genes. 

### 2.7. Differential Expression Analysis

To perform the differential expression analysis, the edgeR package was used [54]. Low-expressed genes were removed, and the remaining genes were normalized using the trimmed mean of M-values (TMM) method [55]. Counts per million (CPM) and log_2_ counts per million (log-CPM) were used for exploratory plots [54] to check the consistency of the replicates. In that sense, mutilBamSummary and PlotCorrelation from deepTools were used to obtain the Pearson correlation of the triplicates: 0.997 on average. Besides, principal component analysis (PCA) and hierarchical clustering of normalized reads per kilobase per million mapped reads (RPKM) were used to get a general overview on the similarity of RNA-sequencing samples [56,57].

DEGs were calculated between LEM2126, LEM3323, LEM5159, LLM2221, LLM2255, LLM2165, LLM2070 compared to LJPC. All genes having a false discovery rate (FDR) value ≤ 0.05 and log_2_ FC ≥ 1 (upregulated) or log_2_ FC ≤ −1 (downregulated) were marked as DEGs. Log_2_ FC was used to evaluate the significance and the change in expression of a gene between both types of samples.

### 2.8. Enrichment Analysis

In order to identify the effects of differential gene expression, a functional enrichment study was carried out using the clusterProfiler Bioconductor package [58]. To this end, DEGs were compared against all the expressed genes in the RNA-seq assay. We obtained Gene Ontology (GO) terms per gene from the TriTrypDB version 56 database, and they were associated to Entrez gene identifiers in an orgDB R object through the AnnotationForge package to be used with clusterProfiler. Therefore, GO enrichment analysis was calculated for biological process ontology terms, defining a statistically enriched term to those having a *p*-adjust value ≤ 0.05.

### 2.9. Protein–Protein Interaction Network Analysis

The DEGs common to all *Leishmania* lines with an FDR value lower than 0.05 were submitted to Search Tool for the Retrieval of Interacting Genes/Proteins (STRING) (https://version-11-5.string-db.org/cgi/network?networkId=bZzuqfjJ3nB4; accessed on 1 May 2022) database version 11.0 [59].

## 3. Results and Discussion

### 3.1. Transcriptomic Profile of L. infantum Lines after Infection of THP-1 Cells

We analyzed the transcriptomic profile of seven *L. infantum* lines isolated from patients suffering from leishmaniasis and experiencing TF (associated or not with drug resistance). Specifically, we examined (i) three clinical isolates (LEM3323, LEM5159m and LEM2126) resistant to Sb, Mil, and PMM, respectively, and (ii) four lines that led to TF (LLM2070, LLM2165, LLM2255, and LLM2221). We compared the transcriptome from these clinical isolates with a genomic reference line (LJPC) at a late timepoint post-infection of THP-1 cells. Those genes having an FDR value ≤ 0.05 and log_2_ FC ≥ 1 (upregulated) or ≤−1 (downregulated) were considered as DEGs.

Our study on the transcriptome of these lines revealed that the number of DEGs varied according to the *Leishmania* line studied; e.g., LLM2070 with 256 genes was the line with the highest number of upregulated genes (Figure 1). On the contrary, the Sb-resistant line LEM3323 presented the largest number of downregulated genes, 307 (Figure 1), among all the lines. LLM2255 had the fewest changes, 84 upregulated and 149 downregulated genes, in contrast with the 402 DEGs from LEM3323. In the remaining lines, the number of upregulated genes ranged from 84 to 222 genes compared to 43–201 downregulated genes (Figure 1). Additionally, we examined the relative number of hypothetical transcripts altered in all the *Leishmania* lines. Hypothetical genes are experimentally uncharacterized genes whose functions cannot be deduced from simple sequence comparisons as they lack sequence similarity with known proteins or domains. The results revealed 30–40% of hypothetical genes among the DEGs in each line. However, the LEM2126 line contained around 50% of the DEGs annotated as hypothetical (Figure 1). Other authors have shown similar values of differentially expressed hypothetical genes in their *Leishmania* transcriptome analyses [27]. One important consideration is that there was no correlation between the DEGs and the corresponding group of *Leishmania* lines (drug-resistant lines or lines that led to TF).

Venn diagrams were used to reveal the DEGs shared by the different *L. infantum* lines, including both the upregulated and downregulated genes. For that purpose, the drug-resistant isolates (Figure 2A) and the TF lines (Figure 2B) were studied separately. A total of 25 DEGs were commonly modulated in the drug-resistant lines LEM3323, LEM5159, and LEM2129 (Figure 2A). Moreover, we found eight commonly modulated genes for the TF lines LLM2070, LLM2165, LLM2255, and LLM2221 (Figure 2B). However, the most significant finding was that the two groups (resistant and TF lines, Figure 2C) shared five DEGs, known as LINF_32000970, LINF_320009800, LINF_200016760, LINF_010013400, and LINF_26SNORNA3. LINF_320009800 and LINF_200016760 both corresponded to hypothetical proteins, while LINF_010013400 coded for a peptidyl dipeptidase. LINF_26SNORNA3 is a small nuclear RNA, i.e., a noncoding RNA located in the nucleolus that is involved in rRNA modification. However, the most important commonly modulated gene was LINF_320009700, which codes for a prostaglandin F synthase, a protein involved in essential lipid metabolism pathways in protozoan parasites [60].

We also drew up volcano plots to compare the fold changes in expression (log_2_) with the corresponding adjusted *p*-values (−log_10_) (Figure 3). The plots represent the gene expression profiles subtracting the statistically nonsignificant genes from the clinical isolates of *L. infantum* lines. The DEGs involved in the most relevant functions linked to TF and based on the bibliography are indicated (Figure 3).

### 3.2. Gene Ontology (GO) Enrichment Analysis of the DEGs

We analyzed the GO-enriched terms of the DEGs from the TriTrypDB version 56 database for the biological process category in clinical isolates of *L. infantum* infecting THP-1 cells at a late timepoint of infection (96 h), focusing on the changes observed in the gene expression of the parasites, to identify common altered processes in *Leishmania*. The individual profiles of each line are shown in Appendix A. This analysis indicated that the drug-resistant parasites LEM2126 and LEM5159 presented a higher number of GO-enriched terms, 28 and 22, respectively (Appendix A). In contrast, TF lines showed a lower number of GO-enriched terms, with zero in the case of LLM2255 and six for LLM2221 (Appendix A).

On the other hand, the analysis of the common GO terms in the different *Leishmania* lines revealed seven GO-enriched categories comprising up- and downregulated genes, as well as hypothetical proteins (Table 1). The GO-enriched terms were those primarily related to macromolecular modification and phosphorylation. Interestingly, the presence of similar pathways reinforces the involvement of their genes in TF/drug resistance. In this way, in *Leishmania* line LEM2126 (a PMM-resistant line), the GO-enriched terms related to macromolecular modification and phosphorylation were due to the upregulation of most of the DEGs; in contrast, in *Leishmania* line LEM3323 (an Sb-resistant line), these GO-enriched terms were due to the downregulation of most of the DEGs (Table 1). In other *Leishmania* lines, like LLM2070 and LLM2221, the GO-enriched terms were due to DEGs’ upregulation or downregulation, respectively (Table 1).

### 3.3. Most Relevant DEGs of L. infantum Lines after Late-Timepoint THP-1 Cell Infection

Chemotherapy is the only effective weapon against leishmaniasis, and this effectiveness is limited by the frequent appearance of drug resistance and TF (1); it is, therefore, crucial to identify potential drug targets able to prevent or reverse such mechanisms. For this purpose, we used *L. infantum* isolates from leishmaniasis patients to study by RNA-seq the transcriptomic changes after late infection in THP-1 cells. Our analysis of the top DEGs in the different *L. infantum* lines after late-timepoint THP-1 cell infection (96 h) showed pathways associated with (i) transport, uptake, and efflux across cellular membranes; (ii) energy metabolism; (iii) redox metabolism and detoxification; and (iv) hypothetical proteins (Table 2).

In the category “transport, uptake, and efflux through cell membranes”, we included several DEGs, such as mitochondrial ornithine transporter 1-like protein LINF_160007200 that was downregulated in the LLM2221 (−1.38 log_2_ FC) line (Table 2). This transporter catalyzes the translocation of ornithine and related substrates, such as arginine, across the inner mitochondrial membrane [61].

Other differentially expressed transcripts identified in this study were LINF_130020800, phospholipid-transporting ATPase 1 which was downregulated with a -1.13 log_2_ FC in the LEM3323 line, and LINF_320010400, a ligand effect modulator 3 (LEM3) family/CDC50 family which was 1.01 log_2_ FC and 2.34 log_2_ FC upregulated in LEM5159 and LEM2126, Mil- and PMM-resistant lines, respectively (Table 2). Phospholipid-transporting ATPase 1 and its beta subunit LdRos3 (CDC50/Lem3) are involved in phospholipid translocation at the plasma membrane of *Leishmania* parasites. Thus, phospholipid-transporting ATPase 1 and LdRos3 form part of the same translocation machinery that determines flippase activity and Mil sensitivity in *Leishmania* [62]. Therefore, changes in expression of the transporter could contribute to the appearance of differences in the uptake of Mil.

Furthermore, the aquaporin LINF_220020300 was overexpressed in the LLM2070 line (1.20 log_2_ FC) (Table 2). Aquaporins are membrane proteins responsible for permeating water, ions, dissolved gases, and other small molecular weight compounds through the protective cell membranes of living organisms [63]. LmAQP1 is an aquaporin from *L. major*, which has proven permeable to glycerol and acts as a drug delivery route for Sb compounds, in addition to playing an important role in osmotaxis [64].

Interestingly, in our study, several transcripts belonging to ATP-binding cassette (ABC) transporters were upregulated in *Leishmania* lines (Table 2). Thus, ABCC1 and ABCC2 (LINF_230007200 and LINF_230007300, respectively) were upregulated in the LEM5159 line (1.04 log_2_ FC and 1.01 log_2_ FC) and ABCA2 (LINF_110018150) was upregulated with a 1.67 log_2_ FC in the LLM2165 line (Table 2). ABC transporters comprise a superfamily of integral membrane proteins involved in the ATP-dependent transport of a variety of molecules across biological membranes, including amino acids, sugars, peptides, lipids, ions, and chemotherapy drugs [65]. They have been associated with drug resistance in various diseases. In *Leishmania*, the first ABC protein identified was MRPA (multidrug resistance protein, PgpA) [54], which is a member of the ABCC subfamily, able to confer resistance to Sb by sequestering thiol–metal conjugates into an intracellular vesicle [14,66]. Meanwhile, the ABCA2 transporter plays a role in phospholipid trafficking, which may modify the vesicular trafficking and the infectivity of *Leishmania* [67]. Thus, overexpression of these ABC transporters could be associated with drug resistance and, as a consequence, TF in these lines.

On the other hand, the category “energy metabolism” also showed differentially expressed transcripts. For example, the argininosuccinate synthase LINF_230007900 was upregulated in all *Leishmania* lines, except LEM3323 and LLM2221, with a log_2_ FC from 1.37 to 2.12 (Table 2). Argininosuccinate synthase is a key enzyme in the urea cycle that catalyzes the rate-limiting step in the conversion of L-citrulline to L-arginine and has a rate-limiting role in high-output nitric oxide synthesis [68,69]. This protein is overexpressed in *L. infantum* axenic antimony-resistant amastigotes and is related to parasite pathogenesis and oxidative stress [70,71].

Another transcript included in the category “energy metabolism” and upregulated (1.43 to 1.43 log_2_ FC) in several *Leishmania* lines was the fatty-acyl-CoA synthetase 2 LINF_010010100 (Table 2). This enzyme catalyzes the formation of fatty-acyl-CoA, occupying a pivotal role in cellular homeostasis, particularly lipid metabolism. Fatty acyl-CoA synthetase is differentially expressed by *L. donovani* amastigotes resistant to Sb [72].

Additionally, the cytochrome c oxidase subunit 10 LINF_230009100 was upregulated (1.02 log_2_ FC) in the LEM5159 line (Table 2). This enzyme is critical for generating the mitochondrial proton gradient for cellular ATP production. It has been observed that Mil inhibits cytochrome c oxidase in *Leishmania* promastigotes [73].

Furthermore, the ATP-dependent phosphofructokinase LINF_290032900 was upregulated (1.44 log_2_ FC) in the LLM2070 line (Table 2). Glycolysis is a central metabolic pathway in all organisms, and a key enzyme of this pathway is 6-phosphofructokinase, which catalyzes the third step of glycolysis: the phosphorylation of fructose 6-phosphate to fructose 1,6-bisphosphate via the conversion of ATP to ADP. This enzyme has been widely studied in *Leishmania* [74,75].

One of the categories studied with several DEGs present in the *Leishmania* lines was that referring to “redox metabolism and detoxification”. A proper intracellular redox balance is vital for the survival and proliferation of all living organisms and depends on reactions based on thiol groups. Most eukaryotic and prokaryotic organisms use glutathione (GSH) or thioredoxin supplemented with the relevant reductases to maintain their reducing capacity. However, the redox metabolism of trypanosomatids is based on the molecule trypanothione, T(SH)_2_. Thus, this molecule would be involved in functions as diverse as DNA replication, defense against oxidizing agents, assembly of iron–sulfur clusters, and detoxification of ketoaldehydes, xenobiotics, and heavy metals.

The gene that codes for the prostaglandin F2α synthase (PGF2S) (LINF_320009700) was found differentially expressed in all the *L. infantum* lines studied, with a log_2_ FC of more than 10 (Table 2). In mammals, prostaglandin synthases catalyze the production of prostaglandins (PGs) using arachidonic acid metabolites as substrates. PGs are lipid mediators involved in cellular processes such as inflammation and tissue homeostasis. PG production is not restricted to multicellular organisms. Trypanosomatids also synthesize several metabolites of arachidonic acid. Nevertheless, their biological role in these early-branching parasites as well as in host–parasite interactions are not well-elucidated. PGF2S has been observed in the *Leishmania braziliensis*-secreted proteome and in *L. donovani* extracellular vesicles; it is an enzyme constitutively expressed throughout promastigote development whose overexpression leads to an increase of infectivity in vitro. Furthermore, a positive correlation between *L. braziliensis* PGF2S expression and pathogenicity in mice has been described [60]. Other researchers have shown PGF2S homologs in *L. major*, *L. tropica, L. donovani, L. infantum, Trypanosoma cruzi*, and *Trypanosoma brucei* [59]. Additionally, it has been described that PGF2S in *T. cruzi* plays critical roles in oxidative stress and susceptibility to benznidazole [76].

On the other hand, tryparedoxin peroxidases (TXNPx) were differentially expressed in the LLM2070 (LINF_290017300), LLM2165 (LINF_260013000), and LEM2126 (LINF_150018800) lines (1.30, 1.58, and 1.20 log_2_ FC, respectively) (Table 2). These are peroxiredoxins belonging to a ubiquitous family of antioxidant enzymes which use redox-active cysteines to reduce their substrates. TXNPxs are broad-spectrum peroxidases which can detoxify hydroperoxides, peroxynitrite, and organic hydroperoxides. Different approaches to trypanosomatids have demonstrated that TXNPxs contribute to parasite survival and the establishment and maintenance of infection [77]. Thioredoxin (Trx) genes are present in the genomes of trypanosomatids and are differentially expressed in LLM2070 (LINF_350008900) and LLM2221 (LINF_260011700) (1.21 and 1.02 log_2_ FC, respectively) (Table 2); however, few studies have been performed on these proteins. The reducing substrate for Trx is T(SH)_2_ [78]. Recently, a new Trx has been described in *T. brucei* (TbTrx2) located in the mitochondria and proposed to be essential for parasite growth in both mammalian and insect stages, as well as relevant for in vivo infectivity [79]. Furthermore, glutaredoxins (Grxs, formerly called thiol transferases) are ubiquitous small thiol–disulfide proteins that catalyze the reduction of proteins that are thiolated by GSH (glutathionylated, PSSG). We found that LINF_010006900 was a DEG in LEM3323 (1.10 log_2_ FC) (Table 2). It is involved in multiple processes, including resistance to oxidative damage, intracellular replication of amastigotes, apoptotic-like cell death, thermotolerance, and proliferation, among other things.

One of the most important genes that codes for gamma-glutamylcysteine synthetase (γ-GCS) (LINF_180022300) (Table 2) was found differentially expressed in LEM5159, LEM2126, and LLM2165 (1.03, 1.65, and −1.04 log_2_ FC, respectively). It is the first enzyme in the glutathione pathway that produces γ-glutamylcysteine, a direct precursor of glutathione [80]. It has been shown to be essential for *L. infantum*, where it confers protection against oxidative stress and Sb^V^.

The relationship of the overexpression of cysteine leucine-rich proteins (LINF_340011100 and LINF_320009800) (Table 2), which our RNA-seq analysis revealed to be in LEM3323 and LEM2126 in the first case and in all the *L. infantum* lines in the second case, with Sb resistance in clinical isolates of *L. donovani* was recently described [81].

Other relevant genes (LINF_050015100, LINF_220010000, LINF_330034900, LINF_080005000, LINF_170016000, LINF_230016800, LINF_290015200, LINF_290007800, LINF_310025400, and LINF_220017900) related to glutathione metabolism as well as the transfer of different chemical groups for detoxification were altered in the various *Leishmania* lines studied in this work (Table 2).

Finally, one gene (LINF_200016760) that codifies for an unspecified protein was differentially expressed (downregulated) in all the clinical *Leishmania* lines with a high log_2_ FC value (Table 2). Future studies focusing on this gene of unknown function, which has such a high level of downregulation, could pave the way to new therapeutic pathways or provide knowledge about drug resistance and TF in leishmaniasis.

### 3.4. Functional and Physical Protein Associations

Based on STRING database version 11.5, we established a functional and physical association map using the only gene (PGF2S) that was differentially expressed (upregulated and with an FDR value lower than 0.05) of the five genes common to all the clinical *Leishmania* lines used in this study that encoded for a protein present in the mentioned database (Figure 4, Table 3).

As described in the previous section, PGs are involved in cellular processes such as inflammation, tissue homeostasis and cellular defense. In this regard, this protein could be considered as an interesting potential drug target taking into account the metabolic functions in which it is involved as well as the overexpression found in the TF *Leishmania* lines studied. In this way, previous studies confirmed the presence of the PGF2S protein in the secretome of *L. braziliensis* and the exosome of *L. donovani*. Additionally, a positive correlation between *Leishmania* PGF2S expression and pathogenicity has been observed in mice [60]. Using the TDR Targets Database (tdrtargets.org), PGF2S was identified as a drug target with a high potential, having a 0.8 druggability index (range: 0.0 to 1.0) [82].

The structure of the LmPGF2S protein has been resolved by crystallization [83], facilitating future studies for drug design using this protein. The structure of this protein could be useful as a template for structure-based drug design efforts for therapeutics that target both *L. major* and *T. cruzi*. However, the high structural similarity of human PGF represents a potential hurdle for the design of selective molecules. Thus, the rational screening of compounds that specifically inhibit the PGF2S could be a way for extending the scarce arsenal of drugs for the treatment of leishmaniasis. One protein intrinsically related to PGF2S is γ-glutamylcysteine synthetase (GSH1) (Figure 4). An increase in GSH1 mRNA levels has been reported in some *Leishmania* samples with in vitro-induced resistance to Sb; it is one of the most significant proteins in the parasite’s redox metabolism [84] and in some Sb-resistant *L. donovani* field isolates [85]. Another protein pinpointed in the interaction analysis, related to redox metabolism, and important for TF in leishmaniasis was putative glutathione-S-transferase/glutaredoxin (described above). From the energetic metabolism category, the analysis highlighted the enzyme known as enolase; this is considered to be an important molecule for parasite survival owing to its role in energy production and, more specifically, in the processes of glycolysis and gluconeogenesis. It also contributes to parasite infectivity, as evidenced by its abundant expression in Sb- and AmB-resistant *Leishmania* parasites [23,86,87], and the protein glycerol-3-phosphate dehydrogenase (glycosomal) (GAPDH). Changes in GAPDH expression may be responsible, at least in part, for the natural resistance to nitric oxide reported in human and canine leishmaniasis [88]. We also located the dihydroxyacetone kinase 1-like protein, which is important for growth and survival and essential for virulence [89].

Lastly, the differential expression pattern of the PGF2S found in all the *Leishmania* lines analyzed at a late timepoint post-infection could have affected the different pathways in the transcriptome of the parasites due to the relationship established between PGF2S and proteins that facilitate the survival and growth of the parasites, partially explaining the TF in the patients infected with these *Leishmania* lines.

## 4. Conclusions

In this work, we evaluated the transcriptomic changes in intracellular amastigotes of *L. infantum* parasites isolated from patients with leishmaniasis and TF (associated, or not, with drug resistance) 96 h after infection of THP-1 cells. The relevant altered functional pathways and genes include transport through cell membranes, energy and redox metabolism, and detoxification. These modifications determine distinct parasite behaviors that are likely to account for parasite survival and TF in patients. Specifically, the gene that codes for the prostaglandin f2α synthase seems to play an important role in pathogenicity and TF since it appears significantly upregulated in all the *L. infantum* lines studied. This gene could be a relevant drug target for leishmaniasis for increasing the efficacy of chemotherapy in cases of leishmaniasis; additionally, its combination with a host-directed therapy including molecules that interfere with certain genes/proteins from the host cells required for the survival of these parasites could represent a new therapeutic strategy for increasing the efficacy of chemotherapy and decreasing TF in patients with leishmaniasis. In general, these results show that at a late timepoint post-infection, the transcriptome of these *Leishmania* lines experiences significant changes that probably contribute to the survival of these parasites in the host cells, as well as to TF. Future works will focus on the proteome of the infecting parasites of *L. infantum* from TF in order to establish a correlation with the transcriptomic data presented in this manuscript; in this way, we will deepen our understanding of the involvement of PGF2S in the TF and their potential as drug targets.

## Figures and Tables

**Figure 1 microorganisms-10-01304-f001:**
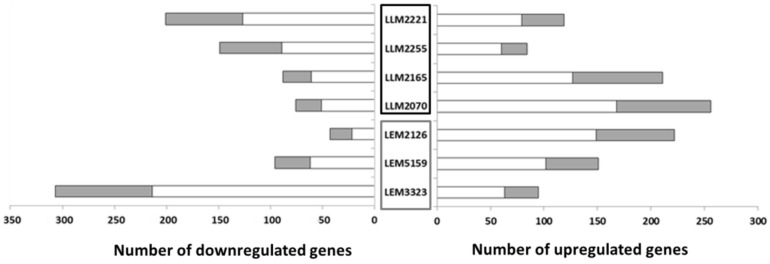
Differentially expressed genes (DEGs) in the *Leishmania* lines after infection of THP-1 cells. The number of DEGs in comparison with the reference line (LJPC) is shown. The gray section of each bar represents the number of DEGs that code for hypothetical proteins. Therapeutic failure lines are included in the black box, drug-resistant lines in the gray box.

**Figure 2 microorganisms-10-01304-f002:**
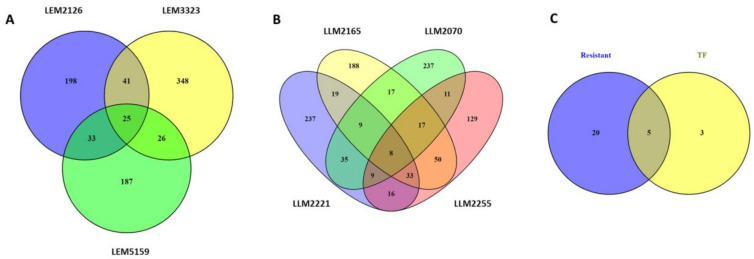
Venn diagrams of differentially expressed genes (DEGs) in *L. infantum* lines after THP-1 infection. Venn diagrams of (**A**) resistant lines, (**B**) therapeutic failure (TF) lines, and the combination of both (**C**). The number of exclusive and common genes in each comparison, including total upregulated and downregulated DEGs, are represented.

**Figure 3 microorganisms-10-01304-f003:**
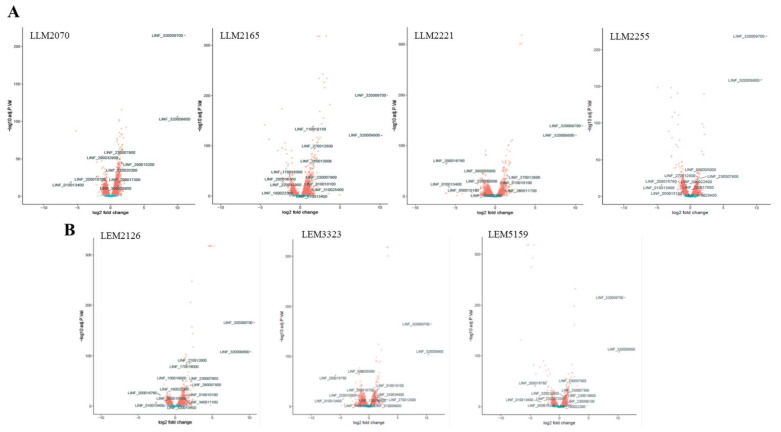
Volcano plots from the RNA-seq data of clinical isolates from the TF (**A**) and drug-resistant (**B**) *L. infantum* lines. The log_2_ FC is plotted on the *x*-axis, and the negative log_10_ FDR (adjusted *p*-value) is plotted on the *y*-axis. Genes are colored according to their FDR: red, FDR ≤ 0.05; blue, FDR ≥ 0.05. Most relevant DEGs based on bibliography and linkage to TF/drug resistance of *L. infantum* lines are shown.

**Figure 4 microorganisms-10-01304-f004:**
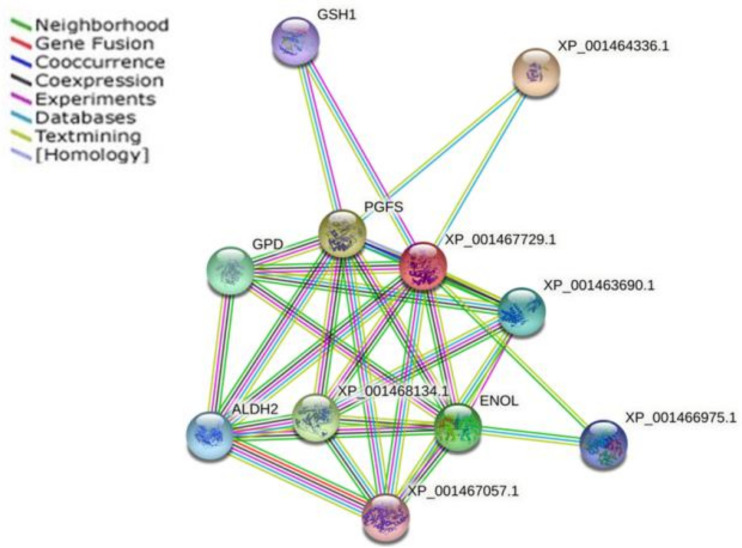
Functional and physical protein associations interaction map of prostaglandin F2α synthase (PGFS). The nodes are representative of protein species, and different line colors show the types of evidence for the association. The STRING tool (http://www.string-db.org; accessed on 1 May 2022) was used to construct the interaction network. The related proteins are listed in Table 3.

**Table 1 microorganisms-10-01304-t001:** Common GO-enriched terms of DEGs for the biological process category in several lines of *L. infantum*.

GO Term	GO ID	Lines	DEGs		
			Up	Down	Hypothetical
Cellular protein modification process	GO:0006464	LEM2126	23	0	0
		LEM3323	7	36	1
		LLM2221	4	17	1
		LLM2070	26	3	0
Protein modification process	GO:0036211	LEM2126	23	0	0
		LEM3323	7	36	1
		LLM2221	4	17	1
		LLM2070	28	4	0
Macromolecular modification	GO:0043412	LEM2126	24	0	0
		LEM3323	9	38	2
		LLM2221	7	21	2
		LLM2070	26	3	0
Protein phosphorylation	GO:0006468	LEM2126	21	0	0
		LEM3323	2	28	0
Phosphorylation	GO:0016310	LEM2126	21	0	0
		LEM3323	2	28	0
Phosphate-containing compound metabolic process	GO:0006796	LEM2126	24	1	0
		LEM3323	2	33	0
Phosphorus metabolic process	GO:0006793	LEM2126	24	1	0
		LEM3323	2	33	0

List of Gene Ontology (GO) biological process enriched terms (adjusted *p*-value < 0.05) of DEGs of different *L. infantum* lines after infection of THP-1 cells. DEGs were selected as follows: log_2_ FC ≥ 1 or log_2_ FC ≤ 1, false discovery rates (FDRs) ≤ 0.05.

**Table 2 microorganisms-10-01304-t002:** Top DEGs in the different *L. infantum* lines after late THP-1 cell infection.

Gene ID	Description	*Leishmania* Line	Log_2_ FC	FDR
1. Transport, uptake, and efflux across cellular membranes				
LINF_160007200	Mitochondrial ornithine transporter 1-like protein	LLM2221	−1.38	4.61 × 10^−9^
LINF_130020800	Phospholipid-transporting ATPase 1-like protein	LEM3323	−1.13	4.33 × 10^−16^
LINF_320010400	LEM3 (ligand-effect modulator 3) family/CDC50 family, putative	LEM5159	1.01	3.13 × 10^−17^
		LEM2126	2.34	0.000207
LINF_220020300	Aquaporin, putative	LLM2070	1.20	1.66 × 10^−31^
LINF_230007200	ATP-binding cassette protein subfamily C, member 1, putative	LEM5159	1.04	3.13 × 10^−17^
LINF_230007300	ATP-binding cassette protein subfamily C, member 2, putative	LEM5159	1.01	4.16 × 10^−21^
LINF_110018150	ATP-binding cassette protein subfamily A, member 2, putative	LLM2165	1.67	4.87 × 10^−140^
2. Energy metabolism				
LINF_230007900	Argininosuccinate synthase, putative	LEM5159	1.79	2.69 × 10^−45^
		LEM2126	2.12	1.61 × 10^−49^
		LLM2070	1.83	1.38 × 10^−63^
		LLM2255	1.60	2.23 × 10^−25^
		LLM2165	1.37	3.22 × 10^−32^
LINF_010010100	Fatty-acyl-CoA synthetase 2, putative	LEM3323	1.43	4.06 × 10^−33^
		LEM2126	1.26	1.59 × 10^−21^
		LLM2221	1.32	4.60 × 10^−21^
		LLM2165	1.07	2.85 × 10^−18^
LINF_230009100	Cytochrome c oxidase subunit 10, putative	LEM5159	1.02	4.96 × 10^−12^
LINF_290032900	ATP-dependent phosphofructokinase	LLM2070	1.44	1.14 × 10^−49^
3. Redox metabolism and detoxification				
LINF_270012800	Amino acid transporter, putative	LEM3323	1.00	1.55 × 10^−20^
		LEM2126	1.90	2.50 × 10^−85^
		LLM2221	1.46	3.52 × 10^−43^
		LLM2255	1.10	1.93 × 10^−27^
		LLM2165	1.75	8.01 × 10^−94^
LINF_290017300	Tryparedoxin-like protein	LLM2070	1.30	1.48 × 10^−18^
LINF_150018800	Tryparedoxin peroxidase	LEM2126	1.10	2.51 × 10^−50^
LINF_260013000	Type II (glutathione peroxidase-like) tryparedoxin peroxidase	LLM2165	1.58	1.00 × 10^−63^
LINF_350008900	Thioredoxin, putative	LLM2070	1.21	1.94 × 10^−15^
LINF_260011700	Thioredoxin, putative	LLM2221	1.01	6.25 × 10^−14^
LINF_010006900	Low-molecular-weight phosphotyrosine protein phosphatase, putative	LEM3323	1.10	9.28 × 10^−19^
LINF_180022300	Gamma-glutamylcysteine synthetase, putative	LEM5159	1.03	4.98 × 10^−10^
		LEM2126	1.65	1.81 × 10^−40^
		LLM2165	−1.04	1.04 × 10^−12^
LINF_340011100	Leucine-rich repeat protein, putative	LEM3323	−1.36	3.02 × 10^−9^
		LEM2126	1.22	3.05 × 10^−16^
LINF_320009800	Leucine-rich repeat, putative	LEM3323	10.25	2.69 × 10^−104^
		LEM5159	10.18	9.34 × 10^−108^
		LEM2126	10.21	8.44 × 10^−109^
		LLM2221	10.14	2.12 × 10^−121^
		LLM2070	10.08	7.91 × 10^−108^
		LLM2255	10.74	4.15 × 10^−160^
		LLM2165	10.26	1.49 × 10^−121^
LINF_050015100	3-mercaptopyruvate sulfurtransferase	LEM5159	−1.25	1.82 × 10^−9^
		LEM2126	−1.24	2.38 × 10^−10^
		LLM2221	−1.76	9.80 × 10^−18^
		LLM2255	−1.06	6.27 × 10^−9^
LINF_220010000	Acetyltransferase (GNAT) family, putative	LEM3323	−1.21	7.79 × 10^−22^
		LLM2221	−1.28	2.13 × 10^−24^
LINF_330034900	Metallopeptidase, clan MF, family M17	LEM3323	1.18	9.33 × 10^−31^
LINF_080005000	Adaptor complex protein (AP) 3 delta subunit 1, putative	LEM3323	−1.79	1.25 × 10^−64^
		LLM2221	−1.52	5.91 × 10^−45^
		LLM2255	1.07	6.32 × 10^−33^
LINF_170016000	META domain containing protein, putative	LEM2126	1.59	2.73 × 10^−85^
		LLM2165	−1.28	1.61 × 10^−42^
LINF_230016800	Alcohol dehydrogenase-zinc-containing-like protein	LEM5159	1.21	1.63 × 10^−22^
LINF_290015200	5-histidylcysteine sulfoxide synthase, putative	LLM2070	1.57	6.10 × 10^−46^
LINF_290007800	D-lactate dehydrogenase-like protein	LEM2126	2.02	6.78 × 10^−42^
LINF_310025400	Acetylornithine deacetylase-like protein	LLM2255	1.06	1.11 × 10^−6^
		LLM2165	1.29	5.34 × 10^−10^
LINF_220017900	ChaC-like protein, putative	LLM2255	−1.46	1.59 × 10^−16^
		LLM2165	−1.50	6.53 × 10^−18^
LINF_320009700	Prostaglandin f synthase, putative	LEM3323	10.85	9.59 × 10^−166^
		LEM5159	11.25	1.29 × 10^−215^
		LEM2126	10.69	8.55 × 10^−167^
		LLM2221	10.95	6.08 × 10^−140^
		LLM2070	11.17	1.74 × 10^−215^
		LLM2255	11.50	4.43 × 10^−219^
		LLM2165	11.24	1.72 × 10^−201^
4. Hypothetical				
LINF_200016760	Unspecified product	LEM3323	−5.77	6.91 × 10^−64^
		LEM5159	−4.08	6.84 × 10^−41^
		LEM2126	−1.99	1.30 × 10^−22^
		LLM2221	−6.16	1.47 × 10^−64^
		LLM2070	−2.02	1.38 × 10^−27^
		LLM2255	−1.73	9.19 × 10^−21^
		LLM2165	−2.11	4.98 × 10^−28^

Profiling of the differentially expressed genes (DEGs) from *L. infantum* lines after late-timepoint infection (96 h) of THP-1 cells as described in the Materials and Methods section. The analysis was based on the log_2_ FC and false discovery rates (FDRs). All the genes presented in this list are statistically significant with an FDR-value ≤ 0.05.

**Table 3 microorganisms-10-01304-t003:** List of proteins directly related to prostaglandin f synthase based on the STRING database.

Protein Code	Description	Relationship Score
XP_001464336.1	Putative glutathione-S-transferase/glutaredoxin	0.919
XP_001467729.1	Putative prostaglandin f synthase (280 aa)	0.803
XP_001468134.1	Putative d-xylulose reductase	0.727
ENOL	Enolase	0.685
GPD	Glycerol-3-phosphate dehydrogenase [NAD+], glycosomal	0.661
XP_001463690.1	Dihydroxyacetone kinase 1-like protein	0.650
ALDH2	Aldehyde dehydrogenase, mitochondrial precursor	0.632
GSH1	Gamma-glutamylcysteine synthetase	0.625
XP_001467057.1	Putative alcohol dehydrogenase	0.575
XP_001466975.1	Putative pyridoxal kinase	0.566

The STRING tool (http://www.string-db.org; accessed on 1 May 2022) was used to construct the network. The relationship score is an indicator of confidence when establishing relationships among proteins. PPI enrichment *p*-value < 0.05.

## Data Availability

The data presented in this study are contained within the article.

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
