# Peer review of "Transcriptome Analysis of Intracellular Amastigotes of Clinical Leishmania infantum Lines from Therapeutic Failure Patients after Infection of Human Macrophages"

_microorganisms, 2022, doi:10.3390/microorganisms10071304_

Round 1

Reviewer 1 Report

The authors are well know members of the Leishmania Scientific community, and the work has been nicely presented, but in my view there are many transcriptome analysis with frequent contradictory results, and with low correlation with  proteome quantitative studies, ranging  in best cases from >20% to 60. Some of this has to do with the fact that not all transcripts end up at the ribosomal translation machinery. So it is always a good idea to try to detect the levels proteins of the most relevant transcripts.

Regarding the prostaglandin alpha  2 synthase, there is not mention to the findings in other leishmania species where its overexpression is associated to parasite drug resistant or fitness and infectivity (Alves-Ferreira et al  13:9 https://doi.org/10.1186/s13071-020-3883-z. or other similar studies. The discussion must be considerable improved to include the comparative analysis with previous studies, the possible role of this enzyme in parasite metabolism, and potential drugs for selective inhibition of the parasite enzyme, otherwise it will add up to the many transcriptome analysis of the pile

Author Response

The authors are well known members of the Leishmania Scientific community, and the work has been nicely presented, but in my view there are many transcriptome analysis with frequent contradictory results, and with low correlation with proteome quantitative studies, ranging in best cases from >20% to 60. Some of this has to do with the fact that not all transcripts end up at the ribosomal translation machinery. So it is always a good idea to try to detect the levels proteins of the most relevant transcripts.

We appreciate and agree with the comment of the referee that it would be very interesting to corroborate the correlation between transcripts and proteins. In fact, our research group will focus on a future project about the analysis of the proteome of infecting parasites of Leishmania infantum from therapeutic failure, in order to establish a relationship with the transcriptomic data presented in this manuscript.

At this moment, it is difficult for us to perform the requested assays since it implies repeating the infections of THP-1 host cells with all the Leishmania lines, at the same conditions previously used to carry out the dual RNA-seq studies and further isolation of the intracellular amastigotes. The method for intracellular amastigotes isolation was labor intensive and required a concomitant analysis of the host macrophage proteins, considering that, as described (Fernando Real et al. Cell Microbiol. 2014; 16(10): 1549–1564), intracellular parasites carries host cell components during its extrusion. Furthermore, commercial antibodies are not available for any of the most relevant proteins that we have highlighted. For some of these proteins, the antibodies need to be requested to other researchers, but in most cases they must be obtained by ourselves. Consequently, although the proposed assay is interesting for us, unfortunately we do not consider it feasible in a reasonable time.

In the revised manuscript, we have added a paragraph in Conclusions section (page 15, lines 501-505) indicating that the proteomic approach will be carried out in future projects.

Regarding the prostaglandin alpha 2 synthase, there is not mention to the findings in other leishmania species where its overexpression is associated to parasite drug resistant or fitness and infectivity (Alves-Ferreira et al 13:9 https://doi.org/10.1186/s13071-020-3883-z) or other similar studies. The discussion must be considerable improved to include the comparative analysis with previous studies, the possible role of this enzyme in parasite metabolism, and potential drugs for selective inhibition of the parasite enzyme, otherwise it will add up to the many transcriptome analysis of the pile

Thanks to the reviewer for her/his comments. We agree with the reviewer’s comment about the relevance of PGF2S and the necessity of improving several aspects in the Discussion section. In this way, in the revised manuscript we have introduced several modifications following the reviewer’s suggestion (Results and Discussion section, pages 12 and 14: lines 377-391 and 492-507, respectively).

Reviewer 2 Report

This is a very straightforward paper, nicely presented. While it will not "solve" Leishmaniasis, it provides useful data for further exploration of Leishmaniasis treatment, and treatment failure.

Author Response

This is a very straightforward paper, nicely presented. While it will not "solve" Leishmaniasis, it provides useful data for further exploration of Leishmaniasis treatment, and treatment failure.

Thanks to the reviewer for her/his consideration and comments.

Reviewer 3 Report

This manuscript describes gene expression analysis to characterize different L. infantum isolates 96h post infection of a human monocyte cell-line. Using differential gene expression analysis, comparing to a reference strain, the study provides information about different Leishmania genes that may be associated with drug resistant or ‘treatment failure’ isolates. The authors conclude specific parasite gene expression changes seen in certain isolates may have some role in drug responses associated with certain isolates.

The writing, structure and presentation of this article needs some serious attention. My critiques are described below:

The manuscript lacks a suitable description of differentiated THP-1 cells and why this is a relevant model to study L. infantum gene expression. Similarly, the study only evaluates Leishmania genes and does not provide any relevant information of the host response, which could be quite different and important to consider with respect to an isolate-specific effect.

The description of the results consistently references ‘DEGs’, which I understand is always comparing to a reference strain in THP-1 cells, but as written it is confusing and is not very clear what is actually being discussed. I would recommend providing some narrative about the experimental conditions up front in the results section and clearly establishing the comparisons (‘i.e., relative to reference strain).

Other than some loose mentions of phenotypes, it is not clear what the difference is between the reference strain and the TF or resistant isolates are. The term ‘therapeutic failure’ requires a brief explanation in terms of both clinical and experimental significance, with references. Similarly, the introduction paragraph (lines 38-48) is awkward and provides inadequate description of drug phenotypes and lacks suitable references. Is the use of semi-colons meant to be concise? More information and better narration are needed, not a list of expression changes. This is often misleading too. For example, in Line 255 it states ‘the transcriptomic changes in L. infantum isolates from leishmaniasis patients’, but should be restated to these are in vitro infections in cells not in patients.

I find Figure 1 uninformative. It could be improved by adding more quality control and showing either similar or distinct host THP-1 responses.

There is no quality control of the data and I am unsure about the technical variation (and thus quality of data). I would like to see information about read depth, signal distribution and sample correlations between all replicates for L. infantum and human expression profiles to understand the variability and data quality.

What is meant by hypothetical genes? This is unclear and needs to be reworded/explained

Line 44 ‘this aspect’ is unclear

Lines 61-63 awkward sentences

Line 64 ‘dual RNA-seq’? what is this?

Line 65 please clarify ‘host’

Line 69 ‘various important factors’ vague, weak and unsubstatiated

Line 70 ‘improving existing expression datasets’ is weak and not very clear

Line 102 remove ‘on the other hand’ please check English style throughout many scientific/English/grammatical problems (i.e., Line 214, ‘we drew up’)

Line 138 annotation file same as genome is unclear, please provide specific annotation information

Data states raw data was generated in triplicate (Line 125) but unclear how triplicates were derived? Please clarify. This information should be presented in the results narrative to support the rigor of the analysis.

Figure 3 poor ascetics and blurry. What are the most relevant DEGs? This seems biased by the authors; how do you know relevance? This is stated several times i.e., line 376 and others.

Table 1: Is the FDR for the enrichment test or DEG definition, what thresholds do these pathways fall under? These pathways are all redundant, are there distinct pathways ? It should be discussed that likely most of the individual pathways in this table contain the same genes, otherwise is misleading

I don’t understand Figure 4 and the results narration is inadequate. Specifically, ‘of the five common to all the clinical Leishmania lines’? makes no sense and is hard to interpret.

The results-discussion could be better grouped to more clearly describe the TF and resistant isolates and how these finding improve our understanding of these concepts. Instead, it reads a long list of different genes and inadequately summarizes these findings.

Author Response

This manuscript describes gene expression analysis to characterize different L. infantum isolates 96h post infection of a human monocyte cell-line. Using differential gene expression analysis, comparing to a reference strain, the study provides information about different Leishmania genes that may be associated with drug resistant or ‘treatment failure’ isolates. The authors conclude specific parasite gene expression changes seen in certain isolates may have some role in drug responses associated with certain isolates.

The writing, structure and presentation of this article needs some serious attention. My critiques are described below:

The manuscript lacks a suitable description of differentiated THP-1 cells and why this is a relevant model to study L. infantum gene expression. Similarly, the study only evaluates Leishmania genes and does not provide any relevant information of the host response, which could be quite different and important to consider with respect to an isolate-specific effect.

Thanks for the reviewer’s suggestion. The human myelomonocytic cell line THP-1 was isolated from a patient with acute monocytic leukemia (Tsuchiya et al. Establishment and characterization of a human acute monocytic leukemia cell line (THP-1), Int. J.Cancer, 1980, 26(2), 171–176). Macrophage-differentiated-THP-1 cells has been considered as a suitable host model for infection of Leishmania lines (1- Dasgupta et al. Infection of human mononuclear phagocytes and macrophage-like THP1 cells with Leishmania donovani results in modulation of expression of a subset of chemokines and a chemokine receptor. Scand J Immunol. 2003, 57: 366–374. pmid:12662300; 2- Jain et al. A parasite rescue and transformation assay for antileishmanial screening against intracellular Leishmania donovani amastigotes in THP1 human acute monocytic leukemia cell line. J Vis Exp. 2012; 70:4054; 3-  Hendrickx et al. In-depth comparison of cell-based methodological approaches to determine drug susceptibility of visceral Leishmania isolates. PLoS Negl Trop Dis. 2019, 13(12): e0007885).  Additionally, these cells have been widely used in the scientific community for in vitro studies of human macrophage functions (Lund ME, To J,et al. The choice of phorbol 12-myristate13-acetate differentiation protocol influences the response of THP-1 macrophages to a pro-inflammatory stimulus. J Immunol Methods. 2016, 430:64–70). Thus, we don’t find necessary to include this description in the manuscript.

Regarding the host response after infection, we have already published host-specific genes that were modulated after the interaction of the THP-1 cells with the clinical isolates of L. infantum with different levels of drug susceptibility to anti-leishmanial drugs associated, or not, to TF (1- Perea-Martínez et al. Transcriptomic Analysis in Human Macrophages Infected with Therapeutic Failure Clinical Isolates of Leishmania infantum. ACS infectious diseases. 2022, 8, 800-810; 2- García-Hernández, et al. New Insights on Drug-Resistant Clinical Isolates of Leishmania infantum-Infected Human Macrophages as Determined by Comparative Transcriptome Analyses. Omics: a journal of integrative biology 2022, 26, 165-177). This information was previously included in Introduction section, lines 91-94.

The description of the results consistently references ‘DEGs’, which I understand is always comparing to a reference strain in THP-1 cells, but as written it is confusing and is not very clear what is actually being discussed. I would recommend providing some narrative about the experimental conditions up front in the results section and clearly establishing the comparisons (‘i.e., relative to reference strain).

Differential expressed genes (DEGs) were calculated between LEM2126, LEM3323, LEM5159 (3 Leishmania infantum clinical isolates resistant to Sb, Mil and PMM, respectively), LLM2221, LLM2255, LLM2165, LLM2070 (4 L. infantum lines that led to therapeutic failure) compared to LJPC (a reference L. infantum line for genomics sequence and for transcriptomic studies). All genes having a False discovery rate (FDR) value ≤ 0.05 and log2FC ≥ 1 (upregulated) or log2FC ≤ -1 (downregulated) were marked as DEGs. Log2FC was used to evaluate the significance and the change in expression of a gene respectively, between both types of samples. This description is included in the Material and Methods section of the manuscript (Section 2.7, lines 185-188) and is applicable to the whole manuscript.

Other than some loose mentions of phenotypes, it is not clear what the difference is between the reference strain and the TF or resistant isolates are. The term ‘therapeutic failure’ requires a brief explanation in terms of both clinical and experimental significance, with references.

The JPC-M5 Leishmania infantum line is a reference line for genomics sequence and for transcriptomic studies that have been employed in several publications (1- González-de la Fuente et al. Resequencing of the Leishmania infantum (strain JPCM5) genome and de novo assembly into 36 contigs. Sci Rep. 2017, 7, 18050; 2- Andrade et al. Comparative transcriptomic analysis of antimony resistant and susceptible Leishmania infantum lines. Parasites Vectors. 2020, 13, 600). In this way, this line is sensitive to all drugs tested (García-Hernández et al. New Insights on Drug-Resistant Clinical Isolates of Leishmania infantum-Infected Human Macrophages as Determined by Comparative Transcriptome Analyses. Omics: a journal of integrative biology 2022, 26, 165-177, doi:10.1089/omi.2021.0185).

As suggested, we have included this information in the Material and Methods section of the revised manuscript (Section 2.2, lines 185-188). The information concerning the clinical isolates of L. infantum is included in the Material and Methods section (Section 2.2, lines 113-121).

Therapeutic failure (TF) has a multifactorial origin, involving a considerable number of factors concerning: (i) the host (immunity or nutritional status), (ii) the parasite (drug resistance, infectivity, parasite localization, accessibility to drugs and coinfection with other pathogens), (iii) the drug (quality, pharmacokinetics) and (iv) the environment (global warming and the expansion of the disease to new geographical areas). Thus, strategies for confronting these drawbacks must include new efficient treatment options. (1- Ponte-Sucre et al. Drug resistance and treatment failure in leishmaniasis: A 21st century challenge. PLoS neglected tropical diseases 2017, 11, e0006052; 2- Vanaerschot et al. Treatment failure in leishmaniasis: drug-resistance or another (epi-) phenotype?.  Expert Rev. Anti Infect. Ther. Early online, 1–10 (2014); 3- Wijnant et al. Tackling Drug Resistance and Other Causes of Treatment Failure in Leishmaniasis. Front. Trop. Dis. 2022, 3:837460). Indeed, our previous manuscripts (1- Perea-Martínez et al. Transcriptomic Analysis in Human Macrophages Infected with Therapeutic Failure Clinical Isolates of Leishmania infantum. ACS infectious diseases 2022, 8, 800-810; 2- García-Hernández et al. New Insights on Drug-Resistant Clinical Isolates of Leishmania infantum-Infected Human Macrophages as Determined by Comparative Transcriptome Analyses. Omics: a journal of integrative biology 2022, 26, 165-177) have shown that modulation of host cells transcripts, after infection with different clinical L. infantum lines, is one of the factors that also contribute to the TF apart from the intrinsic drug resistance of the parasites.

Therapeutic failure (TF) is a clinical phenotype of patients in whom clinical symptoms do not improve after drug treatment (non-response) or reappear after an initial cure (relapse). It has been accepted that in leishmaniasis TF and drug resistance are not necessarily synonymous, being frequently confounded. However, it must be stated that drug resistance to a drug is only one of the possible factors that contribute to TF (Ponte-Sucre et al. Drug resistance and treatment failure in leishmaniasis: A 21st century challenge. PLoS neglected tropical diseases 2017, 11, e0006052).

Following the reviewer’s suggestion, we have defined TF in the Introduction section of the revised manuscript, in order to be clearer for the reader (lines 37-42).

Similarly, the introduction paragraph (lines 38-48) is awkward and provides inadequate description of drug phenotypes and lacks suitable references. Is the use of semi-colons meant to be concise? More information and better narration are needed, not a list of expression changes. This is often misleading too.

Thanks to the reviewer for her/his comments. In the Introduction section we have just described the main phenotypes of resistant Leishmania lines, with some references. Our objective was not to described in detail the main altered phenotypes in the resistant lines, but to show a general overview of the main changes at molecular level in the resistant Leishmania lines. Initially, we have included semi-colons to be concise considering the high amount of information concerning the altered phenotypes in the resistant Leishmania lines.

According with his/her suggestion, we have included more references concerning drug phenotype in resistant Leishmania lines. Also, we have eliminated the semi-colons of this paragraph and we have included new information, avoiding a list of phenotypic changes (Introduction section, lines 43-55). Similarly, we have included more details about other factor that could influence in the therapeutic failure. This information has been included in the revised manuscript (Introduction section, lines 55-59).

For example, in Line 255 it states ‘the transcriptomic changes in L. infantum isolates from leishmaniasis patients’, but should be restated to these are in vitro infections in cells not in patients.

Thanks for the reviewer’s suggestion. We agree with the reviewer’s comment and in the revised manuscript we have rephrased this sentence to specify that it refers to in vitro infections in cells (Results and Discussion, Section 3.3, lines 295-297). 

I find Figure 1 uninformative. It could be improved by adding more quality control and showing either similar or distinct host THP-1 responses.

We appreciate the reviewer’s suggestion. In all the transcriptomic studies, at first we must analyze the number of DEGs in all the lines involved (Leishmania lines in our case). Differences in the number of DEGs (up- and downregulated) between the Leishmania lines could be informative, as a rule this number between DEGs has been described to be similar. However, high changes in DEGs could be related to the adaptation and survival to the intracellular host environment and the escape from the defense of the immune system of patients contributing to the TF.

As we have mentioned, regarding the host response, we have already published host-specific genes that were modulated after the interaction of THP-1 cells with the clinical isolates of L. infantum with different levels of drug susceptibility to anti-leishmanial drugs associated, or not, to TF (1- Perea-Martínez et al. Transcriptomic Analysis in Human Macrophages Infected with Therapeutic Failure Clinical Isolates of Leishmania infantum. ACS infectious diseases 2022, 8, 800-810; 2- García-Hernández et al. New Insights on Drug-Resistant Clinical Isolates of Leishmania infantum-Infected Human Macrophages as Determined by Comparative Transcriptome Analyses. Omics: a journal of integrative biology 2022, 26, 165-177).

There is no quality control of the data and I am unsure about the technical variation (and thus quality of data). I would like to see information about read depth, signal distribution and sample correlations between all replicates for L. infantum and human expression profiles to understand the variability and data quality.

We thank the reviewer for this comment. First, a quality analysis of the sequences is carried out using FastQC. In the report, we observe that most of the sequences (91%) have a quality higher than Q30 and no accumulation of adapters sequences or any other problem. Then, we eliminate those reads that on average have a value lower than Q30 (internally this is performed by samtools) using miARma-seq. After this step, we observe that the size of the library of some samples was larger in comparison with the others. To equal this and to eliminate the possible bias that genes from a sample with a large library appear with higher expression, we use Seqtk software. Finally, we get a set of samples with a similar library size. These sequences are aligned with HISAT2 and then quantified with featureCounts. This last program is parameterized to count only those reads with a minimum mapping quality score >= 10 (in this way, multimapping or misaligned reads are ignored).

Based on this data, a TMM-based normalization is used. Normal and log2 data were used to draw PCAs and MDS-PCoAs to check the reproducibility of the replicates. Unsupervised clustering is also performed to identify any discordant, outlier replicates.

In the following Figures (attached document), we present the MDS and the unsupervised clustering of the data.

All the above information provided show the quality of the data and the analysis performed.

Finally, in order to be clearer for the reader, in the revised manuscript we have modified the text to clarify the analysis performed: Material and Methods, Section 2.6 (lines 164-166 and 171-173), and Section 2.7 (lines 179-180). 

What is meant by hypothetical genes? This is unclear and needs to be reworded/explained

Hypothetical genes are experimentally uncharacterized genes, and their functions cannot be deduced from simple sequence comparisons, as they lack sequence similarity with known proteins or domains. The genome analysis of Leishmania shows considerable number of genes coding for ‘conserved hypothetical’ proteins that are functionally not characterized (Downing et al. Whole genome sequencing of multiple Leishmania donovani clinical isolates provides insights into population structure and mechanisms of drug resistance. Genome Res., 21, 2011, pp. 2143-2156). The computational annotation returns a gene without any homolog in the database and encodes it as “hypothetical‟. Despite several efforts, only 50-60 % of genes have been annotated in the Leishmania genome and their functions are known. The remaining 40% of the genes in any genome is totally unknown in terms of its functions. The experimental characterization of such a huge number of hypothetical genes will take many decades before the biological function encoded by such hypothetical genes is known. Thus, we don’t find necessary to include this description in the manuscript.

Line 44 ‘this aspect’ is unclear

Thanks for the reviewer’s suggestion. We have substituted the word “aspect” instead to “subject” (Introduction section, line 51).

Lines 61-63 awkward sentences

We agree with the reviewer’s comment and in the revised manuscript we have rephrased these sentences in order to be clearer for the reader (Introduction section, lines 81-85).

Line 64 ‘dual RNA-seq’? what is this?

Dual RNA-seq is a well-known technique that provides insights into host-pathogen interactions and is particularly informative for intracellular organisms. Dual RNA-seq quantifies simultaneously RNA transcripts of intracellular pathogens and host cells in a single experiment and can provide insight into both the host and pathogen response to infection.

Line 65 please clarify ‘host’

As we described in the Introduction section, Leishmania are parasites whose life cycle requires them to infect and replicate within host cells as intracellular amastigote forms. In this case, the host cells used were THP-1 cells derived to macrophages. In the references mentioned in this part of the manuscript (1- Perea-Martínez et al. Transcriptomic Analysis in Human Macrophages Infected with Therapeutic Failure Clinical Isolates of Leishmania infantum. ACS infectious diseases 2022, 8, 800-810; 2- García-Hernández et al. New Insights on Drug-Resistant Clinical Isolates of Leishmania infantum-Infected Human Macrophages as Determined by Comparative Transcriptome Analyses. Omics: a journal of integrative biology 2022, 26, 165-177) we focused on the analysis of the transcriptome of the THP-1 cells after infection with the same Leishmania lines used in the actual manuscript. Thus, we don’t find necessary to include this description in the manuscript.

Line 69 ‘various important factors’ vague, weak and unsubstatiated

Thanks for the reviewer’s suggestion. In order to clarify this point of the manuscript we have rephrased this paragraph (Introduction section, lines 96-99).

Line 70 ‘improving existing expression datasets’ is weak and not very clear

Thanks to the reviewer for her/his comment. Following this and the previous suggestions, we have changed the last paragraph of the Introduction section in order to be clearer for the reader (Introduction section, lines 96-99).

Line 102 remove ‘on the other hand’ please check English style throughout many scientific/English/grammatical problems (i.e., Line 214, ‘we drew up’)

We appreciate the comments of the referee about this subject. We have removed from the text the expression “on the other hand” (Materials and Methods, Section 2.3, line 134). Previously to the submission of the original manuscript, we sent the manuscript to a recognized company devoted in translation and revision of scientific works (Trágora; attached a certificate of this revision).

Line 138 annotation file same as genome is unclear, please provide specific annotation information

Thanks to the reviewer for her/his comment. In this section, we wanted to explain that the reference genome (in FASTA format) and the gene annotation (GTF file), were from the same version of TriTrypDB. This line has been deleted as it does not provide valuable information (Results and Discussion, Section 2.6, lines 172-173).

Data states raw data was generated in triplicate (Line 125) but unclear how triplicates were derived? Please clarify. This information should be presented in the results narrative to support the rigor of the analysis.

Thanks to the reviewer for her/his comment. First, we carried to the GENyO facilities the THP-1 cells infected with the different Leishmania lines for the generation of cDNA libraries in three different weeks (independent samples) in order to be accurate and minimize errors. In this way, a cDNA library was generated for each biological sample. In order to be clearer, we have modified this subject in the Material and Methods of the revised manuscript (Section 2.5, line 157).

Figure 3 poor ascetics and blurry. What are the most relevant DEGs? This seems biased by the authors; how do you know relevance? This is stated several times i.e., line 376 and others.

We have performed the figures following the instructions of the journal. All figures are high quality with 300 pp of resolution. Additionally, we have increased the zoom at 350% to check any possible blurriness and we can't find a sensitive loss of definition.

We considered as relevant DEGs those who meet the criteria of False discovery rate (FDR) value ≤ 0.05 and log2FC ≥ 1 (comparison with LJPC), and the function in which they are involved has been published previously and it is linked with therapeutic failure in bibliography. In the revised manuscript, we have introduced a sentence to clarify this point (Result and Discussion, Section 3.1, lines 254-255 and Legend Figure 3).

Table 1: Is the FDR for the enrichment test or DEG definition, what thresholds do these pathways fall under? These pathways are all redundant, are there distinct pathways ? It should be discussed that likely most of the individual pathways in this table contain the same genes, otherwise is misleading

We appreciate the reviewer’s comment. First, the enriched GO terms (pathways) are statistically significant as the adjusted P-value is <0.05. The p-value has been adjusted by the False discovery rate by Benjamini and Hochberg (1995). The legend of the Table has been modified to be more accurate.

As we mentioned in the manuscript, “The individual profile of each line is shown in Supplementary Tables S1-S6.” whereas “the analysis of common GO terms in the different Leishmania lines revealed 7 GO-enriched categories comprising up- and downregulated genes, as well as hypothetical proteins”. The presence of similar pathways in the Table 1 reinforces the involvement of the genes that belong to the mentioned pathways in the TF/drug resistance. We have modified this subject in the Results and Discussion of the revised manuscript (Section 3.2, lines 275-276, 282-284 and Title/Legend of Table 1).

I don’t understand Figure 4 and the results narration is inadequate. Specifically, ‘of the five common to all the clinical Leishmania lines’? makes no sense and is hard to interpret.

We appreciate the reviewer for this observation and we have rephrased this sentence in order to be clearer. The modification can be found in the revised manuscript (Results and Discussion, Section 3.4, line 436).

The results-discussion could be better grouped to more clearly describe the TF and resistant isolates and how these finding improve our understanding of these concepts. Instead, it reads a long list of different genes and inadequately summarizes these findings.

Thanks for the reviewer’s suggestion but we do not agree with the observation. We think that the structure of the manuscript is suitable and accurate. We obtained a huge quantity of information from the transcriptome of 7 Leishmania infantum lines isolated from drug resistance and therapeutic failure after late infection. At the beginning of the manuscript, we separate the DEGs of the resistant lines from those of therapeutic failure. We represented it in a Venn diagram and highlighted the most relevant genes in both groups. After that, we showed in the manuscript the most relevant genes summarizing all the information of the Leishmania lines and grouping them into categories for a better understanding, always pointing out the Leishmania line with changes at the gene level. At this position, we didn’t consider the necessity to separate the Discussion of the genes in different groups (drug resistance and therapeutic failure). The DEGs are explained immediately when they appear in the text, discussing their possible involvement in the TF, but not in a Discussion section apart which could be confusing and repetitive. The flow of the manuscript takes the reader from a big pool of DEGs in 7 Leishmania lines after late infection to a brief list of relevant DEGs and a highlight of PGF2S as a potential drug target.

Round 2

Reviewer 3 Report

Thank you for your responses to my critiques. However, there are still a few points that must be considered:

Yes, THP-1 is a widely used system, but I firmly believe that a single sentence describing and referencing THP-1 to study this pathogen will benefit the overall narrative.

Regarding the host response after infection, Line 90-91 appears to reference growth conditions, not previous host analysis. Are the conditions identical, are these the same data? please clarify.

My recommendation to add description to result section, rather than stuff all the details in the methods, in my opinion, would require the addition of minor narrative sentences and would greatly improve the flow of the article.

A description of hypothetical genes is necessary. Despite being an output of database, it is not scientifically accurate and is confusing.

Author Response

Yes, THP-1 is a widely used system, but I firmly believe that a single sentence describing and referencing THP-1 to study this pathogen will benefit the overall narrative.

Thanks for the reviewer’s suggestion. In the revised manuscript, we have added a sentence describing and referencing THP-1 as host cells for Leishmania infection in order to be clearer and improve the quality of the manuscript (Material and Methods section, lines 106-109).

Regarding the host response after infection, Line 90-91 appears to reference growth conditions, not previous host analysis. Are the conditions identical, are these the same data? please clarify.

We appreciate the reviewer’s comment. In this sentence, we described the line LJPC as a reference Leishmania line. As we mentioned, LJPC has been used in several previous transcriptomic studies reinforcing their use as a reference line. The RNA-seq data obtained from the infection of THP-1 cells with LJPC described in the present manuscript were original, not the same used in the manuscripts referenced. The growth conditions of the Leishmania lines and the THP-1 cells were different too, and these conditions were described in the Material and Methods section (lines 102-106).

My recommendation to add description to result section, rather than stuff all the details in the methods, in my opinion, would require the addition of minor narrative sentences and would greatly improve the flow of the article.

Thanks to the reviewer for her/his comments. As suggested by the reviewer, we have included in the Results and Discussion of the revised manuscript some sentences to improve the flow of the article (section 3.1, lines 193-194; section 3.2, lines 248-249; section 3.4, line 417).

A description of hypothetical genes is necessary. Despite being an output of database, it is not scientifically accurate and is confusing.

Thanks for the reviewer’s suggestion. In the Results and Discussion of the revised manuscript, we have added a sentence about description of hypothetical genes (section 3.1, lines 203-205) in order to be clearer for the reader.